# Systems Approach to Investigate the Role of Fruit and Vegetable Types on Vascular Function in Pre-Hypertensive Participants: Protocol and Baseline Characteristics of a Randomised Crossover Dietary Intervention

**DOI:** 10.3390/nu16172923

**Published:** 2024-09-01

**Authors:** Linda M. Oude Griep, Gary Frost, Elaine Holmes, Nicholas J. Wareham, Paul Elliott

**Affiliations:** 1Medical Research Council (MRC) Epidemiology Unit, Institute of Metabolic Science, University of Cambridge, Cambridge CB2 0QQ, UK; 2MRC Centre for Environment and Health, Department of Epidemiology and Biostatistics, School of Public Health, Imperial College London, London W2 1PG, UK; 3Section of Nutrition, Department of Metabolism, Digestion and Reproduction, Faculty of Medicine, Imperial College London, London SW7 2AZ, UK; 4Centre for Computational and Systems Medicine, Health Futures Institute, Murdoch University, Perth 6150, Australia

**Keywords:** fruit and vegetable consumption, citrus fruit, cruciferous vegetables, dietary assessment, blood pressure, vascular function, metabolomics, gut microbiome, crossover trial

## Abstract

The evidence on the impact of fruits and vegetable types on cardiovascular risk factors remains limited. Specifically, the utilisation of biomarkers to objectively measure dietary compliance and metabolic responses is emerging. This protocol and baseline characteristics of a pilot randomised controlled, crossover, dietary intervention study aimed to examine the effects of citrus fruits, cruciferous vegetables, or common fruits and vegetables on cardiovascular risk factors. A total of 39 volunteers with untreated prehypertension was recruited and consumed a standardised, provided diet with eight daily portions of citrus fruits and cruciferous vegetables, common fruits and vegetables, or a low fruit and vegetable diet (two portions/d, control diet) in a random order for 2 weeks each, separated by a wash-out week. A targeted cohort-based recruitment strategy was utilised and resulted in 74% of participants recruited by re-contacting preselected individuals from two cohort studies with a 15% average enrolment rate. Participants had an average age of 54.4 years (±6.1 years), BMI of 27.9 kg/m^2^, and BP of 135/81 mmHg and were mainly male (67%). The primary outcome was office blood pressure; secondary outcomes included arterial stiffness, lipid profiles, inflammation, cognitive function, and subjective mood. Biofluids, i.e., 24 h urine, stool, and blood samples, were collected for biomarker measurements with multiple metabolomic platforms and the gut microbial composition, together with traditional dietary biomarkers.

## 1. Introduction

Cardiovascular diseases remain the predominant cause of death globally, which is primarily attributable to elevated blood pressure (BP) and suboptimal diet quality, including low fruit and vegetable consumption [1]. Extensive and consistent epidemiological evidence indicates that each 200 g/d of greater fruit and vegetable consumption is associated with an 8% lower cardiovascular disease (CVD) mortality risk [2]. Landmark dietary interventions have demonstrated that the BP-lowering effects of greater fruit and vegetable consumption may explain lower long-term CVD risk [3,4]. This was supported by a meta-analysis of 18 prospective cohort studies that showed a 9 to 11% lower risk of hypertension for those with a high (800 g/d) versus low (40 g/d) fruit and vegetable intake [5]. Research to which types of fruits and vegetables the BP-lowering effects may be attributed, however, has reported inconsistent findings so far [6,7,8,9,10,11,12,13], leaving the underlying mechanisms by which fruit and vegetables may contribute to cardiovascular health largely unexplored.

Fruits and vegetables exhibit biochemical complexity and variations in their nutritional composition across types, which may result in differential metabolic responses. Meta-analyses of prospective cohort studies showed favourable associations between the consumption of apples and pears, citrus fruits, carrots, and tomatoes and CVD risk [2], but significant inverse associations with hypertension were mainly found for apples or pears, citrus fruits, broccoli, and carrots [5]. Acute interventions have demonstrated improved endothelial function following supplementation with orange juice or flavanones [14,15,16], as well as cruciferous vegetables [17,18]. This suggests that certain types of fruit and vegetables may confer greater cardiometabolic benefits.

Traditional self-reported dietary assessment methods in nutritional research are limited by the potential of misreporting [19], which can result in the misinterpretation of research findings. Objective dietary biomarkers measured with greater precision provide higher statistical significance compared to self-reported dietary measures, which will improve the investigation of diet–disease associations [20]. Frequently used established biomarkers of fruit and vegetable consumption include circulating carotenoids and vitamin C [21], but the recent application of metabolic profiling in nutritional research identified new objective dietary biomarkers of certain types of fruits and vegetables. Small-scale acute feeding studies have successfully identified urinary biomarkers of citrus fruit consumption (e.g., proline betaine) [22] and broccoli consumption (e.g., S-methyl-L-cysteine sulfoxide (SMSCO)) [23]. In conjunction with traditional measurement methods, utilising metabolomic strategies using various biofluids and multiple platforms to measure objective biomarkers of citrus fruit and cruciferous vegetables as reliable measures of compliance in a dietary intervention enables a comprehensive investigation of metabolic responses. This approach has the potential to reveal valuable insights into the mechanisms underlying the cardiometabolic benefits of citrus fruit and cruciferous vegetables.

Hence, we present here the protocol of a 9-week randomised, crossover dietary intervention pilot study that employs a systems approach to examine the effect of greater consumption of citrus fruits and cruciferous vegetables and commonly consumed fruits and vegetables on BP and cardiometabolic risk factors in 36 free-living individuals with untreated prehypertension. Additionally, we describe the effectiveness of the recruitment methods and baseline characteristics of enrolled participants.

## 2. Materials and Methods

This study followed the ethical standards of the Declaration of Helsinki and was prospectively registered at clinicaltrials.gov (NCT03410342; https://clinicaltrials.gov/study/NCT03410342?term=NCT03410342&rank=1, accessed on 5 June 2017). All volunteers provided written informed consent prior to screening measurements for participation in this clinical trial. The study received ethical approval from the London Brent Research Ethics Committee (17/LO/0862). The University of Cambridge is the research sponsor for this study.

### 2.1. Objectives

The primary objective is to evaluate the effects of types of plant foods on office systolic and diastolic BP in untreated, pre-hypertensive participants. Secondary objectives include an evaluation of the effects of the intervention on markers of vascular function (arterial stiffness, 24 h ambulatory blood pressure in subsample), biomarkers of cardiovascular risk, i.e., lipid profiles and inflammation, established objective urinary and circulatory markers of the intake of fruits and vegetables, urinary and circulatory metabolic profiles and individual metabolite levels, the composition of the gut microbiota, cognitive function, self-rated general health, subjective mood and sleep, and subjective and objective (in subsample) physical activity.

### 2.2. Trial Design and Ethical Approvals

This is a 9-week pilot dietary intervention study using a randomised, controlled, crossover design. The study consisted of three arms, each involving different types and amounts of fruit and vegetables. The participants’ diets were controlled and provided by the researchers. The intervention periods lasted for 2 weeks each, preceded and separated by one week run-in/washout periods (Figure 1).

Participants attended the clinical research facilities for a screening visit and a total of 6 pre- and post- intervention visits, amounting to 7 visits in total. The study was conducted between November 2017 and May 2018 at the National Institute for Health Research (NIHR) Imperial Clinical Research Facility, located at the Hammersmith Hospital Campus in London. Additionally, study expansion was carried out between January 2020 and June 2021, with intermittent interruptions caused by COVID-19 lockdowns, at the Cambridge Epidemiology and Trials Unit (CETU) situated at the Cambridge Biomedical Campus in Cambridge. Outcome measurements were collected during each of the 6 pre- and post- intervention clinic visits.

### 2.3. Participant Eligibility

We recruited apparently healthy, non-smoking men and women between the ages of 40 and 65 years with high blood pressure (average office systolic BP of 125–160 mmHg) who were not diagnosed with hypertension or using antihypertensive medication. Other inclusion criteria comprised a body mass index (BMI) between 20 and 35 kg/m^2^, a habitual diet low in fruit and vegetables, and living within a reasonable distance from the research sites.

We excluded individuals < 40 or >65 years of age, obesity (BMI ≥ 35 kg/m^2^), an existing or potential diagnosis of hypertension, diabetes mellitus, and cardiovascular, metabolic, renal, liver, thyroid, or gastrointestinal diseases. Additional exclusions were the use of medications that could potentially interfere with energy metabolism, appetite regulation, or a hormonal balance (including inflammatory drugs or steroids, the use of antibiotics, androgens, phenytoin, erythromycin, and thyroid hormones). Current smokers, those with high alcohol intake (>21 or >28 units per week for females and males respectively), excessive moderate to vigorous physical activity (≥10 h per week), adherent to a high fruit and vegetable (≥4.2 portions per day), a vegetarian or vegan diet, users of dietary supplements who were unwilling to pause usage at least 2 weeks prior to enrolment and during the study period, individuals with major food sensitivities, an unstable body weight (weight gain or loss of ≥3 kg in the preceding 3 months), and those with insufficient home food storage at home were also excluded. Women who were pregnant or breastfeeding were ineligible for participation.

### 2.4. Participant Recruitment

A targeted recruitment strategy was exploited using baseline data of the Airwave Health Monitoring Study [24] and the Fenland Study (https://www.isrctn.com/ISRCTN72077169, accessed on 5 June 2017). Cohort-specific teams managed the selection and re-contact of participants. Potentially eligible volunteers were identified from baseline data based on the postcode area, age, systolic BP, absence of a prior diagnosis of hypertension, diabetes mellitus, heart disease, thyroid, kidney, or liver diseases, and no self-reported use of antihypertensive medication. These volunteers had previously consented to be approached for recruitment in subsequent studies.

Invitation letters, approved by the Airwave and Fenland Review Boards, were sent to the identified volunteers. They were asked to respond and provide their contact details via freepost service. In addition, invitation e-mails were sent to volunteers who expressed interest in being contacted for future research studies at the NIHR Imperial Clinical Research Facility. Posters were also circulated at the research facilities, and advertisements were included in staff newsletters.

Positive respondents received the participant information sheet along with a pre-screening questionnaire to assess their initial eligibility. The completed questionnaire was returned via freepost or email or could be completed during a phone call. Respondents who appeared to meet the criteria were invited to a clinic visit for further assessment. Their eligibility was evaluated through a measurement of seated BP, height, weight, waist and hip circumference, and questionnaire information, including the use of alcohol, vitamin supplements, medication, and physical activity (Appendix A). During the screening visit, eligible participants received a diet history interview providing insights into their diet preferences for tailoring of the study diet. Participants were also asked to complete a 3-day food diary before the trial commenced for an evaluation of their habitual diet.

### 2.5. Dietary Intervention

Participants consumed three diets in random order, comprising similar foods but varying in types and quantities of fruits and vegetables. The control diet included ~2 daily portions of fruit and vegetables, while the two intervention diets included ~8 portions of commonly consumed fruits and vegetables or citrus fruit and cruciferous vegetables. Details regarding the types and quantities of fruits and vegetables per intervention arm are presented in Appendix A. The study diet had a targeted macronutrient composition of 49 ± 4% of total energy from carbohydrates, 33 ± 3% from fat, and 17 ± 2% of energy from protein, aligned with the estimated average macronutrient intakes of 1006 participants aged 40 to 65 years from the National Dietary Nutrition Survey (NDNS) [25]. The three study diets comprised similar foods in comparable quantities. To maintain consistent energy levels, the larger quantities of fruit and vegetables were mainly substituted with cereal and snack food items in the control diet. During run-in and washout weeks, all participants received the low-fruit and vegetable diet.

The dietary intervention strategy involved providing participants with weekly groceries, accompanied by general instructions, rotating menu plans, and recipes. The research team ordered the groceries online, which were delivered to the participant’s home by the grocer (Sainsbury’s). Each delivery included foods for all meals, snacks, dairy products, fruits, and vegetables on the menu plan, covering ~90% of the participant’s daily energy requirement.

Participants were assigned to a menu plans with an energy intake level close to their individual estimated daily energy requirement, ranging from 2000 to 3500 kcal/d. To determine the individual energy requirement, the measured basic metabolic rate was adjusted using a physical activity correction factor of 1.4 [26]. To ensure that participants maintained a stable body weight throughout the study, participants were asked to maintain their habitual physical activities. If weight was lost or gained, the participant was assigned to a menu plan with a higher or lower energy intake level accordingly.

Before each delivery, participants received their menu plan with recipes and written instructions. The menu plan outlined foods to be consumed each day. Participants had flexibility to choose from 3 cereal breakfast options, while lunch and dinner meals were fixed. They prepared their meals at home using the provided recipes. To minimise time on food preparation while managing their daily activities, each recipe was designed to yield meal portions for 2 days. Snacks were bought in pre-portioned snack-sized packaging.

The written instructions provided guidance on the study diet, including usual salt and limited coffee consumption to ≤3 daily cups, and of alcoholic beverages to ≤1 daily unit, eating out (limited or menu checked by the research team), no consumption of food products that were not on the menu plan, including other types of fruits and vegetables, no allium vegetables (i.e., onion and garlic), as SMCSO concentrations have been reported to occur in both cruciferous vegetables and allium vegetables [27], not using any other condiments besides those provided, and no dietary supplements, including protein powders and energy drinks.

To encourage adherence to the study diets, participants had some flexibility in their meal consumption. Participants were allowed to consume the daily menu plans, meals, or foods in a different order and to adjust the recipe or preparation method as long as the provided foods were consumed. In addition, participants had the option to consume foods or drinks from a restricted free choice food list, which covered the remaining 10% of their daily energy intake. This list mainly consisted of snacks, desserts, sweetened beverages, and alcoholic beverages (up to a maximum of one unit per day) that did not contain fruits or vegetables. These food items were not included in the food delivery.

### 2.6. Dietary Adherence

To monitor adherence to the study diets, participants were asked to maintain daily food checklists to record the foods and drinks that they consumed and to report any deviations from the menu plan, as well as the use of medication throughout the 9-week study period. Additionally, the measurement of objective biomarkers will be utilised to assess compliance including 24 h urinary potassium and sodium concentrations, circulating carotenoids, and specific metabolomic biomarkers, i.e., proline betaine and SMCSO, indicators of citrus fruit and cruciferous vegetable intake, respectively. Combined self-reported adherence data with objective biomarkers will provide a comprehensive evaluation of participants’ adherence to the study diet.

### 2.7. Intervention Randomisation and Allocation

After the run-in week, eligible volunteers were randomly allocated to one of the six intervention sequences using covariate-adaptive randomisation [28], assigning participants based on sex. We used this randomisation method considering the small sample size, which may lead to random imbalances in sex that could lead to biased results. The method minimises potential imbalances in the allocation of participants by sex across intervention sequences by taking into account the proportional differences of sex within recruitment cohorts and the small sample size of the study.

The randomisation was conducted by an independent statistician who was not involved in any part of this trial using a computer-generated table. Throughout the trial, participants were free to withdraw from the trial at any time by notifying a research staff member. Any data already collected from participants who withdrew would be retained, unless the participant requested its removal. Given the nature of the intervention design and the specific diets involved, neither participants nor the research staff were blinded to the intervention allocations.

### 2.8. Study Measures

For each participant, all measurements at each clinic visit (Appendix A) were taken by the same trained research staff member and were performed in identical sequences according to standardised operating procedures.

#### 2.8.1. Vascular Function Measurements

Two sessions of triplicate office BP and heart rate measurements were taken with an automated oscillometer (Omron HEM-705 CP, Omron Healthcare, Inc., Kyoto, Japan) with at least 30 min in between, for a total of 6 readings. The participant was in a seated position with a sized cuff fitted at heart level. BP readings were taken after ≥5 min of rest, with ≥30 s between readings in a quiet clinical room. The first reading was discarded and the mean of the 2 following readings was calculated. Additionally, a subset of participants was invited to wear a 24 h Ambulatory Blood Pressure Monitor (ABPM, Mobil-O-Graph NG, I.E.M. GmbH, Stolberg, Germany) on the non-dominant arm throughout a 24 h period following each intervention visit, at the same week day. BP readings were captured at regular intervals; every 30 min at daytime (7 am to 11 pm) and every 60 min overnight (11 pm to 7 am).

Arterial stiffness was measured using brachial-femoral pulse wave velocity (PWV, Vicorder, Skidmore Medical, Bristol, UK) during three readings. Participants were in a supine position, angled at ~30 degrees with cuffs placed around the upper arm and upper thigh. Both cuffs were simultaneously inflated, and the corresponding oscillometric signals from each site were captured to automatically calculate the transit time. Measurements were taken once the pressure waveforms were clear and reproducible. To calculate PWV, the distance between the two cuffs was measured for each reading using a measuring tape and was recorded in the Vicorder software (version 8.0).

#### 2.8.2. Anthropometry and Body Composition Assessment

Height was measured without shoes using a wall-mounted SECA 240 stadiometer and was recorded to the nearest 0.1 cm. Body weight and body composition were assessed using bioelectrical impedance analysis with a Tanita analyser (MC-780MA; Tanita Corporation, Tokyo, Japan). Participants stood upright and barefoot and were wearing light clothing during the measurements. BMI was calculated by dividing the participant’s weight (kg) by the square of their height (m). Waist and hip circumferences were measured using fibreglass D-loop tape. Triplicate waist (at the mid-point between the lowest costal region and the iliac crest) and hip (at the greater trochanters) circumferences were recorded to the nearest 0.1 cm.

#### 2.8.3. Cognitive Function Test

Current epidemiological evidence suggests that higher fruit and vegetable consumption may benefit cognitive function [29,30], but data from dietary interventions are sparse. After the physical measurements, participants therefore will undertake a 15 min computerised cognitive function test using a tablet, originally developed for the Airwave Health Monitoring Study [24]. The cognitive assessment consists of six computerised cognitive tests that targets four different cognitive domains: memory (Paired associate learning and Digit span), processing speed (Two choice reaction time), attention (Stroop), and executive function (Numerical and verbal reasoning, Vocabulary).

#### 2.8.4. Questionnaire Data

During the screening visit, participants completed a questionnaire on socio-demographic information, medical history, medication usage, and the use of dietary supplements. At each intervention visit, participants completed a questionnaire related to their general health, physical activity (International Physical Activity Questionnaire (IPAQ) long form) [31], sleep quality (9-item Pittsburgh Sleep Quality Index (PSQI)) [32], appetite profiles (hunger, fullness, satiety and desire to eat on a 100 mm visual analogue scale with “Not at all” and “Extremely” as the anchor points) [33], quality of life (AQoL-8D) [34], mental well-being (7-item Warwick-Edinburgh Scale) [35], and the positive and negative affect schedule (PANAS) [36].

A subsample of participants was invited to continuously monitor their physical activity during each intervention period using an AX3 accelerometer (Axivity Ltd., Newcastle upon Tyne, UK), fitted on the non-dominant wrist by trained research staff. To complement the accelerometer and ABPM data, participants kept a diary to record activities and sleeping times whilst wearing the monitors. These data will be used to objectively evaluate physical activity patterns throughout intervention periods and potential influences on the research findings.

Anonymised data collected on paper were double entered to ensure data quality. Discrepancies were resolved by checking the data on paper by the study coordinator.

#### 2.8.5. Assessment of Habitual Dietary Intake

Participants were asked to complete a 3-day food diary to assess their habitual dietary intake, specifically on 2 week and 1 weekend days in the week after the screening visit. An example of a completed diary was provided, and instructions were given on how to report portion sizes using household measures or reference to the portion-size images provided. Once completed, recorded foods and portions were entered by trained coders from the Nutrition Measurement Platform at the MRC Epidemiology Unit (University of Cambridge, Cambridge) into the dietary assessment system (Diet In Nutrients Out) [37] with incorporated food composition data from the NDNS Nutrient Databank (Year 11, 2018/2019 version by Public Health England).

#### 2.8.6. Sample Collection

Blood plasma and serum, 24 h and spot urine, and if possible, a stool sample were collected at each clinic visit. Participants were provided with sterilised containers and detailed instructions to collect 24 h urine and stool samples in their home environment on the day before the clinic visit. The 24 h urine collection started after the first morning void, recorded as the start time, and finished 24 h later on the morning of the clinic visit. No chemical preservative was added to the urine samples to avoid potential interactions with biomarkers, ensuring minimal degradation of the metabolite composition [38]. The 24 h urine from all containers was pooled, mixed, weighted, and aliquoted.

Participants were asked to collect a stool sample as close in time to the clinic visit as possible using the provided Fecotainer collection kit (AT Medical BV, Zoetermeer, The Netherlands). They recorded the collection time and stored the containers in a refrigerator or in the provided cool bag with freezer elements, which were also used to transport the samples to the clinic visit [39,40]. Upon arrival, the stool sample was homogenised for crude stool aliquots and further processed before faecal water samples were aliquoted.

During each clinic visit, a fasted venous blood sample was drawn using a Vacutainer closed system by trained staff. The filled blood tubes were inverted, kept upright, and brought to the laboratory for immediate processing. Serum samples were kept at room temperature for 20 min before processing, while plasma tubes were immediately centrifuged. Serum and plasma tubes were centrifuged at 3100× *g* for 10 min at 4 °C.

All aliquoted samples were stored at −80 °C for future analysis. A spot urine sample was also collected for dipstick urinalysis and stored for potential future research. Urine dipstick tests were reviewed by the research staff, and any abnormal results were discussed with a physician. If further medical assessment was deemed necessary, participants were notified accordingly.

### 2.9. Laboratory Analysis

Depending on the clinical research facility used, aliquots of fasting serum and 24 h urine samples were sent to either the Clinical Biochemistry laboratory at the Imperial College Healthcare National Health Service Trust (Hammersmith Hospital, London, UK) or the Nutritional Biomarker Laboratory at the MRC Epidemiology Unit (University of Cambridge, Cambridge, UK) for the measurement of lipid profiles, *C*-reactive protein, and 24 h urinary excretion of potassium, sodium, creatinine, and urea. A subset of 20 serum and 24 h urine samples was analysed at both laboratories for cross-validation purposes.

All collected samples of serum, plasma, and 24 h urine will be analysed in a single batch at the Nutritional Biomarker Laboratory for concentrations of carotenoids, vitamin C, lipidomics, and (un)targeted metabolomics using mass spectrometry. In addition, (un)targeted metabolomics using 1^H^ Nuclear Magnetic Resonance (NMR) spectroscopy of serum, 24 h urine, and faecal water samples will be analysed by the National Phenome Center (Imperial College London, London, UK). Collected crude faecal samples will be analysed to determine the composition of the gut microbiota through metagenomic analysis using16S rRNA gene sequencing in collaboration with the Division of Digestive Disease at Imperial College London.

### 2.10. Monitoring

Considering the intervention is based on commonly consumed foods, the risk of harm or adverse events is considered low. Adverse events were recorded, and research staff reviewed each adverse event as it arose according to standardised procedures within each clinical research facility. If necessary, the participant was advised to consult their GP or relevant healthcare professional.

### 2.11. Sample Size Calculation

For this pilot intervention study, we aimed to randomise 36 participants to the trial in order to achieve equal assignment and a substantial number of participants to each of the 6 intervention sequences. This sample size was based on logistic feasibility and practical considerations within the study’s timeframe and available resources, which is in line with recommendations for pilot studies anticipating a medium effect size for a continuous primary outcome in the main trial [41]. Using data from a 3-week crossover dietary intervention on the BP-lowering effects of nutritional advice on a Dietary Approaches to Stop Hypertension fruit and vegetable diet (+4.5 portions/d) versus an average American diet of 2 portions/d of fruit and vegetables in 15 obese hypertensive participants [6], we estimated that a study with 36 participants, a power of 80%, and a two-sided alpha of 5% may detect an effect size of −2.65 mmHg in systolic BP or −3.06 with a power of 90% (https://hedwig.mgh.harvard.edu/sample_size/, accessed on 6 February 2017).

### 2.12. Statistical Analyses

To evaluate the effectiveness of the research recruitment from different sources, we calculated the response rate, screening visit completion rate, and enrolment rate. The response rate was calculated specifically for the cohort-based recruitment strategy by dividing the number of received positive responses by the number of individuals who were contacted. The screening visit completion rate and enrolment rate were calculated for each recruitment source individually by dividing the number of completed screening visits or the number of enrolled eligible individuals by the number of individuals who responded positively from that particular source.

The main analyses will be conducted by an independent statistician using the latest version of Stata (StataCorp LLC, Lakeway, TX, USA). Descriptive statistics will be used to report the mean ± standard deviation (SD) for each outcome variable and will be reported at each clinic visit, as well as the change between the start and the end of each intervention period. Continuous variables will be presented as the mean ± SD, non-normally distributed continuous variables as the median and interquartile range, and categorical variables as the frequency and percentage. The normality of outcome variable distributions will be assessed graphically. If not satisfied, appropriate transformations will be applied to achieve normality.

All participants with baseline and at least one outcome measure at the end of one of any intervention period will be included in the analysis. A linear mixed effects regression model will be employed to estimate the effect of the two dietary interventions with the control group, along with a 95% confidence interval. The intervention group and period will be set as fixed effects, and the participant identifier as a random effect. Considering the small sample size and to limit complexity of the models, we will consider the random intercept in the analysis to account for individual variability. If the analysis indicates the need for the addition of a random slope to the model, we will run this as a sensitivity analysis. This approach adheres to the CONSORT recommendations for crossover trials, as it avoids an adjustment for baseline values of outcome variables [42,43]. Since this study is designed as a pilot study, no adjustment for multiple comparisons will be made. The results will be reported as effect sizes with corresponding 95% confidence intervals.

## 3. Results

### 3.1. Recruitment Response Rates

In total, 330 adults expressed positive interest in the study (Figure 2) with 74% recruited from the cohort-based strategy (Table 1). Following completion of the pre-screening short questionnaire, 109 participants completed the screening visit and 39 participants were enrolled to the study. This resulted in an average screening visit completion rate of 33% and an average enrolment rate of 12% (Table 1). Postal mailings to pre-selected participants from the Fenland and Airwave cohort studies had the highest average screening visit completion rate (39%) and average enrolment rate (15%), compared to 16% and 3%, respectively, for the advertisement-based recruitment strategy.

### 3.2. Baseline Characteristics of Enrolled Participants

Out of 42 participants who started the run-in week, three participants discontinued during run-in and 39 participants were randomised to one of the six intervention sequences (Figure 2), 21 participants at Imperial College London and 18 participants at the CETU in Cambridge. Participants aged 41 to 65 years (mean 54.4 years), of whom 33% were female and 87% were from a white ethnic background, took part in the study (Table 2)**.** The majority of participants were (self-) employed (77%), completed a college degree (41%), and never smoked (67%), and alcohol consumption was reported by 92% of men and 69% of women. Office BP was on average 135/81 mmHg with 41% of participants having high-normal BP and 31% stage 1 hypertension. The average BMI was 27.9 (SD 3.2) kg m^−2^, with 54% of participants being overweight (BMI ≥ 25.0 kg m^−2^) and 28% obese (BMI ≥ 30.0 kg m^−2^). As a measure of central adiposity, 49% of participants had a waist circumference higher than 88 cm and 102 cm for women and men, respectively.

The median (IQR) self-reported physical activity was 3546 (1308–7148) MET-minutes per week. Although the median self-reported daily sitting time was 5.4 h, 60% of participants were categorised as highly physically active according to the IPAQ guidelines [31].

Out of the 39 participants, 3-day food diaries were completed and analysed for 33 participants (Table 3). The median intake of total energy was 2144 (1937–2669) and 1777 (1350–2060) kcal for men and women, respectively, with overall carbohydrates contributing 51% of total energy, 34% from fat, and 16% from protein. The median daily intake of fruit was 116 g (62–220), 159 g (97–179) of vegetables, and 96 g (55–158) of meat. There were eight participants (21%) who reported adherence to a special diet prior to start of the trail, including mainly using dairy-free or low-fat food products.

## 4. Discussion

We presented here the study design, protocol, recruitment results and characteristics of enrolled participants of a crossover dietary intervention on the effects of types of fruit and vegetable consumption on vascular function. Groceries, including fruits and vegetables, were delivered to the participant’s home weekly to improve compliance. Dietary compliance throughout the study will be measured based on objective dietary biomarkers, including established and recently discovered biomarkers of fruit and vegetable consumption and specific types, i.e., citrus fruits and cruciferous vegetables. As a proof-of-principle of a systems approach, we will utilise metabolomic phenotyping using various biofluids and measured with different platforms for a comprehensive investigation of the potential underlying mechanisms of cardiometabolic responses following high fruit and vegetable consumption.

The population of this dietary intervention comprises middle aged, free-living men and women with elevated office BP who are not using anti-hypertensive medication with the majority experiencing high-normal BP or stage 1 hypertension. With an elevated BP combined with apparent central obesity, a sedentary lifestyle, and a lower diet quality, these participants have a moderate-to-high risk of developing CVD later in life for which the adoption of nonpharmacological healthy lifestyle changes may prevent the development of CVD [44].

Previous large intervention studies that reported challenges recruiting individuals with elevated BP showed that targeted mass mailings were the most effective recruitment strategy [45]. In line with these findings, we found that targeted cohort-based recruitment resulted in higher recruitment yields compared to traditional methods. The postal mailings to approach and pre-screen potential volunteers was the most feasible method to recruit participants for this small-scale study, but main limitations, including hands-on administration and postal costs, a longer turnaround time, lack of capacity to send a reminder, and potential selection bias, may affect the generalisability of research. Recent developments in using digital tools for recruitment and research study management may overcome these issues. Our findings support the advantage of utilising cohort data as a targeted recruitment strategy, while future research may probe digital tools to improve the effectiveness of research recruitment.

The application of the objective measurement of dietary biomarkers in dietary intervention studies is emerging as monitoring dietary compliance based on subjective dietary assessments, i.e., food checklists are challenging due to misreporting related to their self-reported nature and lack of detail [19]. Serum carotenoids and plasma vitamin C are established concentration markers of fruit and vegetables that are commonly used in dietary intervention studies [21]. These markers combined have demonstrated the better prediction of overall fruit and vegetable intake in a 4-week highly controlled parallel randomised controlled feeding trial with 30 healthy volunteers randomised to either 2, 5, or 8 portions per day than considered as a single biomarker [46]. Fruit and vegetables however are complex as they possess a wide diversity of bioactive compounds with varying concentrations influenced by environmental factors, including storage and processing and inter-individual variation in digestion, absorption, and bioavailability, which may impact biomarker responses and thus vascular function [47]. Besides established biomarkers, i.e., lycopene for tomato and β-carotene for carrot intake [21], we will utilise metabolomics for a comprehensive measurement of recently discovered biomarkers of specific types of fruits and vegetables, i.e., proline betaine as marker of citrus fruit [22] and SMSCO as a marker of cruciferous vegetables [23], to monitor compliance with the consumption of fruit and vegetable types. Additionally, the (un)targeted metabolomics of blood, urine, and faecal samples before and after each intervention period allows for the potential discovery of metabolites and a comprehensive investigation of potential synergistic metabolic responses to fruit and vegetable consumption that may provide insights into the underlying mechanisms of cardiometabolic health for future larger dietary intervention studies in this area.

### Strengths and Limitations

The main strength of this study is its crossover design to effectively deal with inter-individual variation as each participant serves as their own control with biofluids collected at the start and end of each of the intervention period for measurements of biomarkers and health outcomes. Although situated in a free-living environment, the provision of the intervention diet keeps the background diet relatively stable to limit the intake of fruit and vegetable types that are not part of the intervention or potential food sources containing similar biomarkers that may bias the results. The main limitation of this intervention study includes the small number of enrolled participants, which may be sufficient for a pilot study but limits the detection of significant effect sizes on office BP. In addition, office BP measurements are subject to substantial variability. To limit the variability in office BP, we used a standardised protocol, and the first reading was discarded to calculate the mean of two subsequent readings [44]. We also invited a subset of participants to wear a 24 h ABPM device to provide multiple and more reproducible BP readings throughout the day in the usual environment of the participant [48]. These BP measurements combined with additional measures of vascular function and markers of CVD will enable the identification of potential underlying mechanisms of fruit and vegetable consumption on BP. Another limitation is that the majority of the recruited participants were male and had a white ethnic background. Considering the well-known gender-based differences in dietary habits and adherence, motivation and lifestyle factors, the execution of this dietary intervention and in particular adherence to a predefined menu plan with specified diet restrictions may be challenging. This may limit the anticipated results and the generalisability of results to the current UK or other populations with greater ethnical diversity.

Other limitations relate to the nutritional nature of the study, including variation in the provided portions, i.e., the weight of a broccoli head or apples can significantly vary, unavailability of certain foods due to seasonality or limited stock, and lack of control over consumed portion sizes. Unavailable foods will be substituted with comparable foods and provided in the weekly delivery, or alternative solutions will be discussed with the participant.

## 5. Conclusions

This randomised, crossover pilot study employs a systems approach utilising metabolomic strategies to provide insights into the underlying metabolic mechanisms of fruit and vegetable consumption and specific types on cardiometabolic health benefits. The findings of this research may contribute to the development of future, larger dietary intervention studies.

## Figures and Tables

**Figure 1 nutrients-16-02923-f001:**
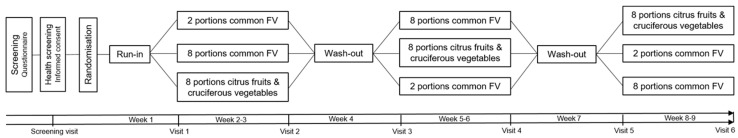
Schematic overview of the randomised dietary intervention with a cross-over study design.

**Figure 2 nutrients-16-02923-f002:**
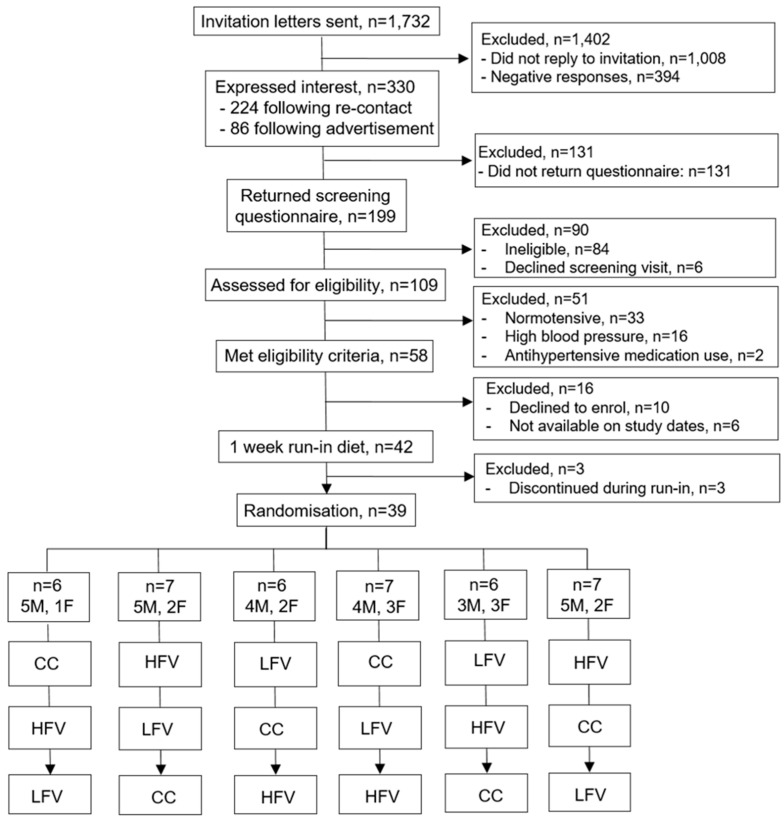
Flowchart of participant recruitment, eligibility, and enrolment.

**Table 1 nutrients-16-02923-t001:** Responses by research recruitment strategy and source.

	Positive Responses, *n* (%)	Response Rate ^1^, %	Screening Visits, *n*	Screening Visit Completion Rate ^2^, %	Eligible Participants, *n*	Enrolled Participants, *n*	Enrolment Rate ^2^, %
Re-contact pre-selected participants cohort studies	244 (73.9)	14.1	95	38.9	53	36	14.8
Airwave study	125 (37.9)	9.2	33	26.4	25	19	15.2
Fenland study	119 (36.1)	32.2	62	52.1	28	17	14.3
Advertisement	86 (26.1)	NA	14	16.3	5	3	3.5
CRF volunteer database	68 (20.6)		10	14.7	4	2	2.9
Poster/advertisement	18 (5.5)		4	22.2	1	1	5.6
Total	330		109	33.0	58	39	11.8

^1^ The response rate was calculated by dividing the number of individuals that returned a positive response by the number of individuals contacted. ^2^ The screening visit completion rate and enrolment rate were calculated for each recruitment source by dividing the number of individuals that completed the screening visit or enrolled, respectively, by the number of individuals that returned a positive response from that source.

**Table 2 nutrients-16-02923-t002:** Characteristics of 39 enrolled study participants at the screening visit ^1^.

	Total (*n* = 39)	Males (*n* = 26)	Females (*n* = 13)
Age, y	54.4 ± 6.1	53.0 ± 5.3	57.2 ± 7.0
Age range, y	41–65	42–61	41–65
Male, *n* (%)	26 (67)	26 (100)	13 (0)
Ethnicity, *n* (%)			
White	34 (87)	22 (85)	12 (92)
Black Caribbean	3 (8)	3 (11)	0 (0)
Other	2 (5)	1 (4)	1 (8)
Blood pressure, mmHg			
Systolic	135.1 ± 6.8	135.9 ± 7.1	133.6 ± 6.2
Diastolic	81.0 ± 6.8	80.9 ± 7.0	81.0 ± 6.8
Heart rate, bpm	69.6 ± 12.6	67.7 ± 12.3	73.2 ± 13.0
Blood pressure category ^2^, *n* (%)			
Normal	11 (28)	7 (27)	4 (31)
High-normal	16 (41)	10 (38)	6 (46)
Stage 1 hypertension	12 (31)	9 (34)	3 (23)
Anthropometrics			
BMI ^3^, kg/m^−2^	27.9 ± 3.2	28.0 ± 3.3	27.7 ± 3.2
Waist circumference, cm	96.9 ± 9.5	99.7 ± 9.2	91.3 ± 7.5
Hip circumference, cm	107.0 ± 6.0	106.6 ± 6.0	107.6 ± 6.3
Waist-to-hip ratio	0.91 ± 0.07	0.93 ± 0.06	0.85 ± 0.05
Body fat, %	27.4 ± 6.8	24.3 ± 5.2	33.6 ± 5.3
Basic Metabolic Rate, kcal	1803 ± 304	1965 ± 212	1481 ± 172
BMI categories ^4^, *n* (%)			
Healthy weight	7 (18)	5 (19)	2 (15)
Overweight	21 (54)	13 (50)	8 (62)
Obese	11 (28)	8 (31)	3 (23)
Central obesity ^5^, *n* (%)	19 (49)	9 (35)	10 (77)
Physical activity, *median (IQR)*			
Total activity, MET-min/week ^6^	3546 (1308–7148)	4167 (1661–7289)	3210 (981–4320)
Sitting time, hrs/day	5.4 (4.4–9.7)	5.0 (4.2–9.6)	7.6 (5.0–10.0)
Physical activity level ^6^, *n* (%)			
High	21 (60)	15 (63)	6 (55)
Moderate	12 (34)	8 (33)	4 (36)
Low	2 (6)	1 (4)	1 (9)
Educational level, *n* (%)			
College or university degree	16 (41)	9 (35)	7 (54)
O levels/GCSEs	12 (31)	6 (23)	0 (0)
A levels	6 (15)	10 (38)	2 (15)
Other	5 (13)	1 (4)	4 (31)
Employment, *n* (%)			
(Self-) employed	30 (77)	20 (77)	10 (77)
Retired	9 (23)	6 (23)	3 (23)
Smoking status ^7^, *n* (%)			
Never smoked	26 (67)	17 (65)	9 (69)
Past smoker	11 (28)	8 (31)	3 (23)
Missing	2 (5)	1( 4)	1 (8)
Alcohol consumer, *n* (%)	33 (85)	24 (92)	9 (69)
Special diet, *n* (%)	8 (21)	5 (19)	3 (23)
Low fat	4 (10)	1 (4)	3 (23)
Diary free	2 (5)	2 (8)	0 (0)
Other	2 (5)	2 (8)	0 (0)

^1^ Data are generally presented as the mean (standard deviation, SD) or number of participants (%), except for physical activity data, which are presented as medians (interquartile ranges). ^2^ Blood pressure categories were defined following the classification of office BP of the 2023 European Society of Hypertension guidelines [44]: normal, systolic BP of 120–129 mmHg and diastolic BP of 80–84 mmHg; high-normal, systolic BP of 130–139 mmHg and diastolic BP of 85–89 mmHg; Grade 1 hypertension, systolic BP of 140–159 mmHg and/or diastolic BP of 85–89 mmHg. ^3^ BMI was calculated as [weight (kg)/height (m)^−2^]. ^4^ BMI cut-off points for weight status were a healthy weight, 18.5–24.9 kg/m^−2^, overweight, 25.0–29.9 kg/m^−2^, and obese, >30.0 kg/m^−2^. A BMI > 35 was an exclusion criteria for the study. ^5^ Central obesity was defined by the waist circumference using sex-specific cut-off points (females ≥ 88 cm, males ≥ 102 cm). ^6^ MET, metabolic equivalent of task. Following IPAQ guidelines for data processing [31], we truncated the activity duration to 180 min, excluded participants from this analysis (*n* = 4) who reported a total sum of time variables of >960 min, and calculated MET-minutes per week for each activity, which was used to categorise participants into high, moderate, and low physical activity levels using defined cut-offs. ^7^ Current smoking was an exclusion criterion for the study.

**Table 3 nutrients-16-02923-t003:** Habitual dietary intakes estimated using 3-day food diaries during the screening of 33 enrolled study participants and by gender ^1,2^.

	Total (*n* = 33)	Males (*n* = 22)	Females (*n* = 11)
Energy, kcal	2035 (1762–2462)	2144 (1937–2669)	1777 (1350–2060)
Carbohydrates, en %	51.0 (47.3–53.6)	50.3 (44.8–53.6)	52.5 (48.7–55.3)
Mono- and disaccharides, en %	18.7 (14.3–21.6)	17.7 (14.3–20.4)	21.5 (14.7–23.2)
Fibre, g/1000 kcal	10.9 (9.2–12.4)	10.7 (8.8–11.7)	11.6 (9.3–12.9)
Alcohol, en %	0 (0–2.8)	0 (0–5.4)	0 (0–2.8)
Protein, en %	16.4 (13.5–19.5)	16.5 (13.8–20.0)	14.4 (13.1–17.9)
Fat, en %	34.1 (29.6–38.7)	34.3 (28.9–38.5)	33.4 (30.2–39.9)
Monounsaturated fat	12.5 (11.1–13.7)	12.9 (11.1–13.7)	12.5 (2.8–11.8)
Polyunsaturated fat	5.7 (4.4–6.4)	5.6 (4.4–6.3)	5.8 (4.4–7.4)
Saturated fat	12.1 (10.7–15.2)	12.2 (11.0–15.0)	11.6 (10.1–16.0)
Fruit, g/d	116 (62–220)	127 (60–222)	90 (62–220)
Vegetables, g/d	159 (97–179)	155 (110–179)	159 (83–179)
Meat, g/d	96 (55–158)	139 (76–167)	55 (46–106)
Fish and shellfish, g/d	1 (0–42)	13 (0–46)	0 (0–42)

^1^ Completed 3-day food diaries were received from 33 out of 39 enrolled participants (22 males, 11 females). ^2^ Data are presented as medians (interquartile ranges).

## Data Availability

Anonymised data may be available in the future for collaborative research. Data requests must be authorised by the principal investigator (L.M.O.G.) and may need additional approval of the Research Ethics Committee.

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
