# Peer review of "Systems Approach to Investigate the Role of Fruit and Vegetable Types on Vascular Function in Pre-Hypertensive Participants: Protocol and Baseline Characteristics of a Randomised Crossover Dietary Intervention"

_nutrients, 2024, doi:10.3390/nu16172923_

Round 1

Reviewer 1 Report

Comments and Suggestions for Authors

The manuscript is important because it is framed within the concept of open science, where it is also important to publish the protocols of the research studies, in this particular case it is an intervention study with a crossover design.

1.     Line 43 . What does it mean  FV?

2.     In general, the first time that an acronym is mentioned it should be explained eg SBP,  etc

3. table 2 uses Body Mass Index consistently, but at other times, BMI is used. Try to use in the same way in the table.

4.     In addition  to ClinicalTrials.gov, Was the clinical trial registered in another clinical trial registry like those of (,  EU Clinical Trials Register, WHO, or similar)If the answer is yes, please provide the registration information

5.     Another question is the sample size and power calculation in lines 358 -359. How did you get to this: “A power of 80% and a two-sided alpha of 5% may detect 358 an effect size of -2.65 mmHg in systolic BP, or -3.06 with a power of 90%.” Did you use software or some mathematical formula? Please explain and include the references.

6.     The information about the clinical trial registration is included at the end of the paper in the section of ethicas aspects. Please include also a reference to the registration in the material and method section of the paper; provide also in brackets  a direct link to the registration

7. Please explain with more detail covariate adaptive randomization in so no expert can understand it.  Adaptive clinical trials are relatively recent.ç

8. In some parts of the manuscript, it is written in the past tense, while in others, it is in the future tense (for example, the statistical analysis part). It would be interesting if the tense used in the writing is the same throughout the article.

Comments on the Quality of English Language

See the above comments.

 In some parts of the manuscript, it is written in the past tense, while in others, it is in the future tense (for example, the statistical analysis part). It would be interesting if the tense used in the writing is the same throughout the article

 I have reviewed the manuscript and I think it could be published after solving several minor issues,

Reviewer 2 Report

Comments and Suggestions for Authors

The authors conducted a randomized crossover trial aiming to examine the effects of citrus fruits, cruciferous vegetables, or common fruits and vegetables on blood pressure and cardiometabolic risk factors in adults with untreated prehypertension. A total of 39 adults were finally recruited and randomized. In this manuscript, the authors report the trial design, participants' recruitment, and participants' baseline characteristics. This trial is essential in light of the global health burden of cardiovascular diseases. Also, the results would improve our understanding of the health benefits of fruits and vegetables. The manuscript was, in general, clearly written. There are some comments.

1.     Materials and Methods (2.8. Study measures): I recommend specifying the primary and secondary outcome measures in the first paragraph.

2.     Materials and Methods (2.12. Statistical analyses): A linear mixed model will be used. Please describe the possible covariance structures between random effects. In addition, individual variability was considered as a random effect, which could be a random intercept, random slope, or both. Please describe the possible random effects at the participant level (random intercept, random slope, or both?) that would be considered in the planned analysis.

3.     Discussion (Lines 478-487): "To investigate the effects of types of fruits and vegetable consumption on vascular function independently of body weight, we aim to -. --" This paragraph seems to describe the study protocol. I recommend moving it to "Materials and Methods."

4.     Discussion: This trial was planned to recruit both men and women (Line 113). However, of the 39 adults who were finally randomized, 26 were men, and only 13 (33%) were women (Table 2). I recommend a further discussion of this result. For instance, would this contribute to the challenge in executing the planned dietary intervention among free-living adults (particularly men), for which the biomarker-based dietary adherence monitoring would have potential value? Would this possibly affect the generalizability of the trial results? Also, among the 39 adults, 34 (87%) were of white ethnic backgrounds (Table 2). Would this possibly affect the generalizability of the trial results? A discussion is recommended.
